# Networks and Emotions in Cooperative Work: A Quasi-Experimental Study in University Nursing and Computer Engineering Students

**DOI:** 10.3390/healthcare8030220

**Published:** 2020-07-20

**Authors:** Pilar Marqués-Sánchez, Isaías García-Rodríguez, José Alberto Benítez-Andrades, Iván Fulgueiras-Carril, Patricia Fernández-Sierra, Elena Fernández-Martínez

**Affiliations:** 1SALBIS Research Group, Faculty of Health Sciences, Campus of Ponferrada, University of León, 24401 Ponferrada, Spain; pilar.marques@unileon.es (P.M.-S.); elena.fernandez@unileon.es (E.F.-M.); 2SECOMUCI Research Group, Escuela de Ingenierías Industrial e Informática, Universidad de León, Campus de Vegazana s/n, 24071 León, Spain; isaias.garcia@unileon.es; 3SALBIS Research Group, Department of Electric, Systems and Automatics Engineering, University of León, 24071 León, Spain; 4Linde Group España, Calle Hamburgo N 16, 24401 Ponferrada, Spain; ivafulcar95@gmail.com; 5Department of Nursing and Physiotherapy Health Science School, University of León, Avenida Astorga s/n, Ponferrada, 24401 León, Spain; pferns03@estudiantes.unileon.es

**Keywords:** emotions, social network analysis, nurse students, engineering students, interdisciplinary learning, cooperative work

## Abstract

University students establish networks that impact on their behavior. Social Network Analysis (SNA) allows us to analyze the reticular structures formed and find patterns of interaction between university students. The main objective of this study was to observe the impact of interdisciplinary collaborative work between nursing and computer engineering students on the collaboration and friendship networks, emotions and performance of the participants. It is a quasi-experimental descriptive study with pre- and post-intervention measurements. The contact networks analyzed showed an increase in density in the post-intervention period. The most central people in the network corresponded with those who considered positive emotions most in their academic environment, while the less central people coincided with those who highlighted negative emotions. Academic performance was only significantly associated in the collaboration network, between this and OutdegreeN. This study shows the impact of interdisciplinary activities on teaching methodologies and the repercussions of emotions on curricular activity.

## 1. Introduction

University students establish social relationships with significant behavioral consequences [1]. Schoolrooms are suitable environments for establishing social network structures in which information is exchanged, leading to increased effectiveness in problem solving, understanding concepts or exchanging opinions [2]. In this context of network and students, the evidence suggests that the position that each student in these interaction networks is related to their academic performance [3]. The importance of these relationships is reinforced by the increase in team activities that are developed through the new teaching methodologies [4].

Human Relations Theory stresses that the central nexus of any organization is subdivided into two parts, the human and the interactive [5]. Both terms are interrelated within any organization, that is, human behavior will depend on the interconnections established between the members of a social group and on their well-being, derived from the connections established with the environment and from social norms [6]. This framework is of particular interest to this research because of the impact connections have on relationships.

On the other hand, and within the relational context, the emotions that arise in university students could predict their academic coping strategies, their willingness to learn [7] and their best performance [8]. In particular, the emotional intelligence of students becomes especially important as they are exposed to challenging interpersonal situations in contexts unknown to them [9]. Therefore, the relationships established in the academic environment could also influence such personal aspects as emotions, social support and friendship of the students. The literature has verified that the degree of friendship that arises from the contacts between students could generate positive emotions, while the inability to relate to others and loneliness, on the other hand, would generate negative emotions [10].

Although the learning process involves cognitive, emotional and social aspects, there is little research that analyzes the emotions referred to by university students in relation to different teaching strategies used during their training [11], since at university the teaching of disciplinary content has prevailed with little consideration of the emotions generated. However, more and more authors are pointing out the fundamental role that emotions play in the learning–teaching process [12,13].

When academic activities generate positive emotions students are more motivated, they pay more attention, feel in control of their learning process, work harder and are more committed to their studies [14,15]. This makes it easier for students to feel confident and develop relevant learning [16,17]. On the other hand, when students generate negative emotions, students experience frustration, maladjustment and dropout, limiting their ability to learn [18,19].

To understand the meaning of relationships that are established within the classroom, a theoretical framework that studies these links is necessary. The framework that covers the study of relationships is called Social Network Analysis (SNA). The SNA allows the analysis of the different constructs at the level of the reticular structure, a distinctive feature of this method of analysis [20]. Its objectives focus on finding patterns of interaction between actors within the social context, and empirically evaluating the links established [21]. One of the most relevant patterns refers to the centrality of a subject, which analyzes and describes the position that each individual occupies within the network, that is, the relationships that a node has directly or indirectly with the rest of the actors [22]. Their interest is key since it has been found that an individual’s position can influence academic performance [3] and the transfer of ideas and knowledge [4]. Identifying the most central actors means finding the most influential subjects within the network [4]. The SNA framework provides us with the ability to analyze whether the most central individuals within an academic context are also those who get the best grades, which was demonstrated in a study with Master’s Degree in Business Administration (MBA) students [23]. Previous experiences have shown that relationships between university students could be enriched by interdisciplinary interventions that complete their learning process, and this would be reflected in their performance [24]. It has been found that these interventions could enhance communication, facilitate learning and creativity and improve problem solving and critical thinking [25]. Furthermore, there is evidence that heterogeneity in work teams promote the generation of new ideas that favor innovation [26]. According to the exposed, it could be of interest to carry out studies in which relationships between collaborative networks of students from different disciplines and academic performance can be explored.

As regards the study of interdisciplinary social structure and the optimization of relational links, the literature already provides some previous experiences. Some examples are studies on the exchange of inter-professional knowledge on electronic records related to drug issues [27], collaboration between different professionals in the creation of green spaces in large cities [28] or collaboration between groups of scientific research [29], among others. However, there is an acute lack of articles in the field of university education that analyze the relationships in interventions that include students from different disciplines in depth, an aspect of interest for a working environment which requires more and more collaboration between different professionals.

This study is part of an Educational Innovation project carried out at a Spanish public university. The students selected were studying to different degrees: Nursing and Computer Engineering. They had to carry out a cooperative work in which the nursing students presented a demand or need, and the engineering area had to provide a technological proposal for that demand. Previous studies in the United States have already pointed out that, to improve the effectiveness of healthcare, the use of electronic technology will be essential [30]. This is also shared by the group of teachers, who understand that, in order for future professionals to work collaboratively, it is necessary to start with previous experiences from the university classrooms.

For all these reasons, the following research question was proposed: “How do interdisciplinary networks influence individual academic performance?” To answer this question, the following objectives were set:Analyze the structure of student networks before and after the interdisciplinary intervention.Identify the degree of similarity of the students with respect to the emotions perceived in the academic environment.Study the relationship of interdisciplinary networks of university students with academic performance.

## 2. Material and Methods

A quasi-experimental study was developed with pre- and post-intervention measurements without a control group.

### 2.1. Sample Description

The selection of the sample was made by means of a convenience sampling. The sample was made up of 50 students from two different degrees at a public university in Spain: 26 students were in the third year of their Nursing degree and 24 students were in the fourth year of their Computer Engineering degree (Table 1). The degrees were studied at different campuses 113 km apart.

### 2.2. Variables

The variables collected were:Descriptive variables: Sex (male and female) and university degree (Degree in Nursing and Degree in Computer Engineering).Network structural variables:
-Centrality. Position in the network: IndegreeN (degree of relationships received by the individual), OutDegreeN (degree of relationships issued), EingvectorN (degree of prestige or influence) and BetweennessN (degree of intermediation) [31].-Density. Number of relationships present divided by the number of possible relationships [21].Emotional variables: Happiness, joy, love, anger, fear and sadness. These were selected based on Bisquerra’s theory of emotions [32], which highlights six main emotions: three positive, namely joy, love and happiness, and three negative, namely fear, anger and sadness.Academic performance: Academic evaluation of the results of the cooperative work.

### 2.3. Instruments Uses to Collect Data

The questionnaires were digitized and presented as online forms, so that they could be accessed via a URL on any computer or laptop, mobile device or tablet by entering a username and password. The server in which they were hosted had SSL encryption, which allowed for data security and they travelled via secure HTTPS (Hypertext Transfer Protocol Secure) without compromising the privacy of the users.

The questionnaire collated the following variables:Degree (Engineering or Nursing).Sex.To measure the structural variables of centrality, a Likert-type scale from 0 to 4 points was used to assess the sociocentric networks of the entire list of participants in the study. The networks valued were: (a) friendship network: Which of the following partners do you consider to be friends? [33]; and (b) collaboration network: Which of the following colleagues would you ask for help when a problem/doubt/difficulty arises in the academic field? [34].When quantifying the emotional variables in the academic environment, two parameters were used: (a) “the intensity with which students consider that the following emotions are present in their academic environment”; and (b) “the intensity with which students consider that the following emotions should be present in their academic environment”. A Likert-type scale was used to answer them, with 0–4 being weighted for the emotions: “happiness”, “love”, “joy”, “anger”, “fear” and “sadness”, with 0 being the “lowest intensity” and 4 the “highest intensity”.For the measurement of the academic performance, the final grade obtained in the work was used, scored from 0 to 10. This grade included the individual grades of the written work and the oral presentation.

### 2.4. Procedure

Data were collected in two stages: the first was in the first face-to-face session (March 2018) and the second was after the presentations of the teamwork (June 2018).

The Excel program was used for the treatment of the descriptive, emotional and performance variables. In the case of the measures of centrality of the sociocentric networks, square matrices were created for each of the collaboration and friendship networks. Subsequently, data were dichotomized by means of intermediate coding (Table 2). All the centrality values were standardized, according to version 6.622 of the UCINET program [35].

### 2.5. Intervention

An interdisciplinary intervention was implemented in which the participants were grouped into nine teams made up of students from both degrees. They were to carry out cooperative work on technologies applied to the health field. The members of each group were selected by the professors. The students from the different degrees did not know each other before the intervention, so it was decided to carry out the first session in an outdoor space, to facilitate a relaxed atmosphere among them.

The beginning of the intervention was a 5-h session in which the the methodology and objectives of the work were explained to the students, creating a first face-to-face contact between them. They were given a period of 40 days to develop the contact through online tools. The nursing students would identify a healthcare problem whose solution was to be found by the engineering students through technology applied to healthcare. Once this was done, all of the participants presented the work via video conference as a link between the two campuses. The engineering students presented the healthcare need and the nursing students the technological solution.

At the end of the assignment, the teachers graded all the students individually. The final grade of the work consisted of a weighted average of the written work and the oral presentation.

### 2.6. Ethical Considerations

Student participation in the study was voluntary. They were informed of the objective of the study and could leave it at any time. The confidentiality and anonymity of the subjects was maintained at all times during data collection and analysis using the tool described by Benítez et al. (2017), which generates simulated names for the participants [36].

The Helsinki Declaration, Law 14/2007 of 3 July on Biomedical Research, as well as the legal rules on data confidentiality (Regulation EU2016/679 and Constitutional Law 3/2018 of 5 December on the Protection of Personal Data and Guarantee of Digital Rights) were followed. This study was approved by the Ethics Committee of the University where the study was conducted (Ref. ETICA-ULE-026-2018), which guaranteed compliance with all ethical and legal issues.

### 2.7. Data Analysis

All statistical analyses were performed using SPSS v. 25.0. Pearson’s correlation coefficient for parametric values and Spearman’s correlation coefficient for non-parametric values were used to determine the correlations, establishing a degree of significance of p<0.05 and p<0.01.

For the structural analysis, specialized software from UCINET (version 6.622) was used [35]. The analyses focused on measures of centrality and density in one-mode networks (same set of actors). The analyses of students and emotions were carried out using two-mode networks (one set of individuals and another set of emotions).

## 3. Results

The aim of this study was to analyze the relationship between the friendship, collaboration networks and emotions with academic performance, through a collaborative work intervention between university students of nursing and engineering. To this end, the results are divided into the following points: descriptive sample results, pre- and post-intervention network results, network position and emotions results and network position and academic performance results.

### 3.1. Descriptive Results on the Sample

The sample was made up of a total of 26 students in the Nursing degree, 85% of whom were women and 15% men, and 24 students in the Computer Engineering degree, 21% of whom were women and 79% men. Thus, in the Nursing degree there is a predominance of women, while in the Computer Engineering degree there were more men.

### 3.2. Results of Pre- and Post-Intervention Networks

In Table 3, the density of the four networks that were constructed from the data collected is shown. The results show a large difference in the density of the pre- and post-intervention networks. The Friendship had values of 0.074 and 0.105 pre- and post-intervention, respectively, and the collaboration network of 0.053 pre-intervention and 0.107 post-intervention.

The qualitative analysis was calculated by visualizing the friendship network and the collaboration network, pre- and post-intervention.

In the pre-intervention networks (Figure 1), few connections between nursing and computer engineering students can be observed. The vast majority of relationships were established between participants belonging to the same degree. With respect to gender, the results are inconclusive since the percentages in both grades are not comparable.

On the other hand, the networks show a slight difference in the overall way the two networks behave. In the friendship network (Figure 1a), there is a greater separation according to the university degree being studied. A small group of students act as bridges between both sub-networks of university students. In the collaboration network (Figure 1b), the nucleus of students who act as bridges is greater, with nursing students mainly occupying this position. In this sense, the network shows a greater robustness and difficulty in splitting up. Figure 2 shows an abstraction of this description.

As regards the post-intervention description, both the friendship network (Figure 3a) and the collaboration network (Figure 3b) showed clear dynamism following the intervention carried out through cooperative work, given that they increased the number of contacts between students of both degrees. That is, the post-intervention networks increased the density of connections. In the graphs, a slight tendency to grouping according to university degree can be observed, although less so than in the pre-intervention networks.

### 3.3. Networks and Emotions

Student networks and emotions were analyzed, including positive emotions (joy, happiness and love) and negative emotions (anger, fear and sadness). The results were obtained by applying Mode Two network analysis, because two sets of nodes, students and emotions were collected. To interpret the perception of emotions in our research properly, two items were raised: the emotions that are present and those that should be present in cooperative work. Each item generated a network. Next, Table 4 and Table 5 detail the percentage of Nursing Degree students and Computer Engineering Degree students and how they perceived the emotions in each of the stages of the intervention. Table 4 describes the pre-intervention emotions, noting that the greatest difference between students is in the love and fear emotion.

Table 5 refers to the post-intervention stage. In the emotions perceived by the students as present in the cooperative work, there are differences in the emotions love, fear and sadness. On the other hand, in those they consider should be present in the work, the biggest differences are reflected in the love emotion, and in all the negative emotions, anger, fear and sadness.

In an attempt to summarize the results, a qualitative description of the networks and emotions in the post-intervention stage has been carried out (Figure 4a). In the case of the network of emotions that are present (Figure 4a) in the cooperative work, most of the students expressed that positive emotions were present in their academic environment, although three engineering students and one nursing student highlighted the presence of negative emotions. As regards the emotions that should be present (Figure 4b), it can be observed that the students mostly opt for positive emotions. A small core of students selected both positive and negative emotions, but not exclusively negative ones.

### 3.4. Position of Students in the Network and Emotions (In Cooperative Work)

Based on these results, a qualitative study was analyzed regarding the students’ position in the network and emotions at the end of the cooperative work intervention, including emotion as an attribute. The emotions selected for this result were happiness and anger, since both obtained a higher degree of response among positive and negative emotions, respectively. The network under scrutiny was the friendship network, since it represents a more selective type of relationship. The position in the network was analyzed using the normalized degree of centrality, the DegreeN (Figure 5a). In the graphical representation, a higher degree of DegreeN is represented by the size of the node. It can be seen that there is a relationship between centrality and the presence of happiness in university students. The most central nodes valued that happiness was present in their academic environment, and only some less central nodes considered that it was not present.

As regards the Anger emotion, individuals with higher DegreeN consider that it is not present in cooperative work (Figure 5b). In general, students who consider Anger to be present in cooperative work are in more peripheral positions and with less DegreeN. Only three Nursing Degree students in central positions, i.e., with high DegreeN, highlight the importance of this emotion.

### 3.5. Student Network Position and Academic Performance Results (In Cooperative Work)

An analysis was carried out between the student’s position in the network and academic performance. The position was analyzed in the friendship network and collaboration network through the following centrality measures: OutDegreeN, InDegreeN, DegreeN, BetweenN and Eigenvector. Our findings showed the correlation only between the collaboration network for OutDegreeN and academic performance, being r=−0.402, in which the correlation is significant for p<0.01.

The study presented analyzed the data resulting from an educational intervention. This intervention consisted of an interdisciplinary cooperative work implemented among university students studying computer engineering and nursing. The mixed work teams had to identify a health problem and propose a technological solution. The proposed task was evaluated as part of the Nursing Care Management course qualification in the case of nursing students and Semantic Modeling in the case of engineering students.

Data were collected pre- and post-intervention, which allowed changes in friendship and collaboration networks to be detected as a result of the intervention. An analysis of the emotions considered by the students in their academic environment was carried out, as well as the correlation of the centrality variables and the performance of the participants. One of the main results obtained was an increase in the density of friendship and collaboration networks in the post-intervention. On the other hand, the emotions perceived did not show significant changes between the two stages. However, a relationship was found between the centrality of the student in the network, and the emotions perceived by them. As regards the correlations with performance, only the relationship of performance with OutdegreeN in the collaboration network was obtained as a statistically significant result.

In this sense, the findings should be evaluated cautiously given that it seems logical that the number of contacts always increases after having to carry out collaborative work. In addition, shared emotions do not always have to be positive since sometimes tensions are generated between students when making decisions regarding their collaborative work. However, this intervention provides new evidence regarding the importance of networks between students, and possibly the tensions originated between them, could also have been perceived as favorable. In other words, some tension between team members could be motivating. As regards interdisciplinary interventions between the health and technological fields, interest has been shown on numerous occasions: monitoring of the diabetic patient [37], repair and reproduction of human cells and tissues by 3D bio-printing [38] or the implementation of telemedicine as a measure to reduce infections in the COVID-19 pandemic [39]. In reference to interventions carried out in the university educational field, evidence has been found of interdisciplinary work among students in the technological and economic fields for the development of Apps [40], and between students of Biotechnology and students of Business Administration to evaluate the economic viability of innovative projects in the change of biotechnology [41]. Furthermore, the study by Pawar et al. in 2017 concluded that multidisciplinary simulation education offers a positive emotional response from students and prepares them for relevant learning, dynamic thinking and a willingness to continue learning [42]. However, no evidence has been found where the study population are future nurses and computer engineers. The new technologies in the health field have numerous applications and their demand is increasing, but they often do not go on to healthcare practice [43] and interdisciplinary collaborative learning facilitates students’ acquisition of planning skills [41].

Under these premises, the SNA has been incorporated as a methodology for studying the relational pattern, bringing a distinctive approach to the work. The intervention facilitated the creation of new contacts between the students, since the task they had to carry out would be evaluated later, and the participation of the members of both degrees was essential to carry it out and obtain a good qualification. For this reason, the SNA was introduced to monitor the networks that were established, given that there is evidence that this approach is appropriate for the analysis of relationships in collaborative work [44]. Our findings are in line with previous studies [1], since they have enabled us to assess the structure of contacts and their influence on learning, useful information for teachers as a strategy in the learning–teaching process.

Direct comparison of our results with other studies has not been easy, since no other have been found that take into consideration the same variables in this type of intervention. We can highlight the significant increase in contacts in both the friendship network and the collaboration network, as a consequence of the exchange of information, opinions and knowledge among the participants. In line with these results, other interventions have also led to an increase in the density of the final networks with respect to the initial networks [45]. This increase in post-intervention contacts may suggest that the passage of time strengthens relationships and increases the density of the network. However, in a study by Liébana-Presa et al. (2018) into student nurse networks, the density of collaboration and friendship networks in the first year had higher levels than the density of networks in more advanced courses, possibly because the insecurities that arise in the first year led to increased relationships [46]. Of interest are the contributions made by Zsófa Boda (2020) in relation to his studies with engineering students recently admitted to the university. After the intervention, a higher rate of friendship between students assigned to the same working group than between pairs of students who were not, was detected. However, these effects gradually diminished as the academic year progressed, since, as they were induced relationships, they had less stability [47].

On the other hand, the visualization of subnets in the graphic representations should be emphasized, since most of the relationships were established between students from the same degree course, possibly because they share similar aspects. This fact is in line with the concept of homophily, which is highly regarded in the SNA. Homophily is understood as the tendency of relationships between people with similar social characteristics, “beyond what is expected under conditions of randomness” [48]. Therefore, our study shows similar results to previous research that found how professionals with similar fields of interest or belonging to the same branch of knowledge are more likely to establish relationships [49].

In reference to the emotions perceived by the participants, we discovered the direct relationship between centrality and positive emotions, while negative emotions were related in an inversely proportional way. In this sense, Morelli et al. (2017) determined that people with high levels of well-being are characterized as being more central in networks, although in their case they found no relationship between centrality and negative emotions [50]. Other experiences argue that positive emotions in students favor their adaptation, and facilitate the creation of social relationships and friends [10]. Specifically, our findings are aligned with studies of positive emotions in nursing students, which found a direct relationship between positive emotions and wellbeing, coping and perceived competencies [51].

On the other hand, the scientific literature argues that engineering students have negative emotions to a greater extent than science students [52]. For these reasons, it is important to introduce teaching methodologies related to emotional intelligence into both degrees. Moh et al. (2020) emphasized that those responsible for training future nurses must consider that mood affects learning and knowledge transfer, adding that having knowledge about how to deal with feelings may help them in a future career [53]. Similarly, engineering teachers should integrate the development of emotional skills into their methodologies, as this would impact on academic performance and the impact could be extended into the workplace [54].

In relation to the results obtained after correlating the centrality variables and performance, we found previous experiences with different results. In one of them, the academic performance is not influenced by the friendship relationships of the university students, that is, the relationship lacks significance [55]. However, in another intervention, the students who showed more central positions in the network were those who achieved higher grades, so social interactions had an impact on their academic results [23]. Similar results were obtained when analyzing the collaborative network of first- and second-year classrooms of school in the north of the Netherlands, where a positive relationship between the achievement and the position in the network was clearly demonstrated [56].

In this regard, the academic performance variable used in this work could have some bias, since it is not the student’s final performance in the subject. However, it has been correlated with a performance measure that represents a group evaluation, since we understand that this is the coherent measure to relate to collective network behavior.

The study has a number of limitations, since certain variables that could facilitate the understanding of the results were not taken into account. On the other hand, it is possible that a greater number of face-to-face sessions would have facilitated the relationships between those course participants who did not belong to the same working group or to the same degree, which would have led to an enrichment of the results. In the present study, students reported the presence of certain emotions during collaborative work, but, in future studies, it would be convenient to evaluate these emotions during situations in which students are subjected to different levels of pressure. Finally, the intervention was not planned with a control group in order to compare the data obtained and observe possible differences. However, the work includes a series of novel contributions by covering interdisciplinary teaching methodologies in the university environment, analyzing the contacts formed and relating them to emotions and performance.

## 4. Conclusions

This study includes an interdisciplinary intervention with nursing and engineering students. The intervention consisted of collaborative work between students from both degree courses. Pre- and post-intervention networks and emotions were measured, as well as their relationship with performance. The most relevant conclusions are listed below:The interdisciplinary intervention modified the relational pattern of the nursing and computer engineering students, increasing the number of relationships in the collaboration and friendship networks.The graphic representation of the pre- and post-intervention networks shows the tendency of the participants to relate to aspects of homophilia based on the university degree.The emotions perceived by the students did not show significant changes after the intervention. A direct relationship was found between the students’ centrality and positive emotions and an indirect relationship between centrality and negative emotions.There is a significant relationship between OutdegreeN in the collaboration network and academic performance.The results show the benefits of introducing interdisciplinary activities in teaching methodologies, so that several technological proposals for healthcare demands were achieved.

## Figures and Tables

**Figure 1 healthcare-08-00220-f001:**
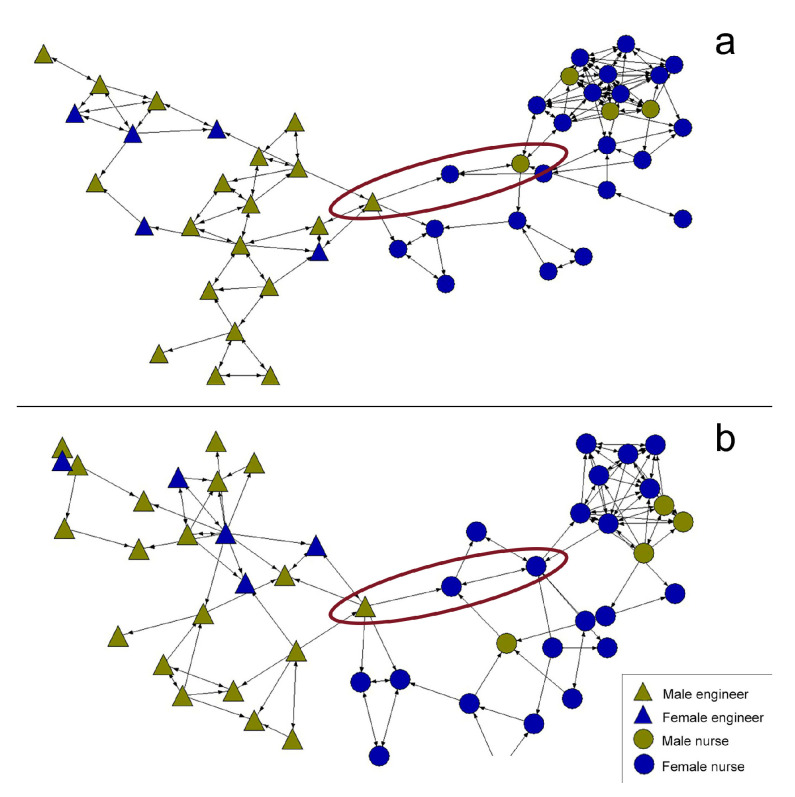
Pre-intervention friendship network (**a**) and collaboration network (**b**).

**Figure 2 healthcare-08-00220-f002:**
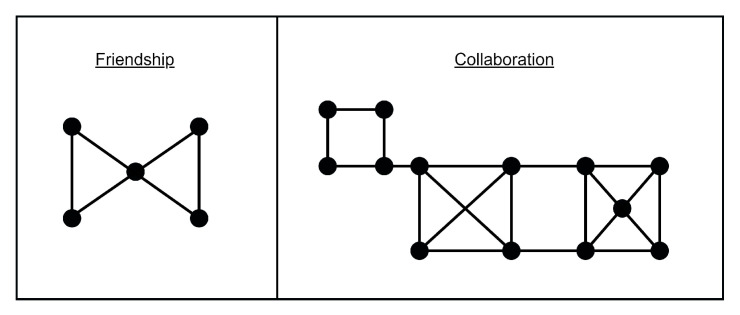
Pre-intervention scheme.

**Figure 3 healthcare-08-00220-f003:**
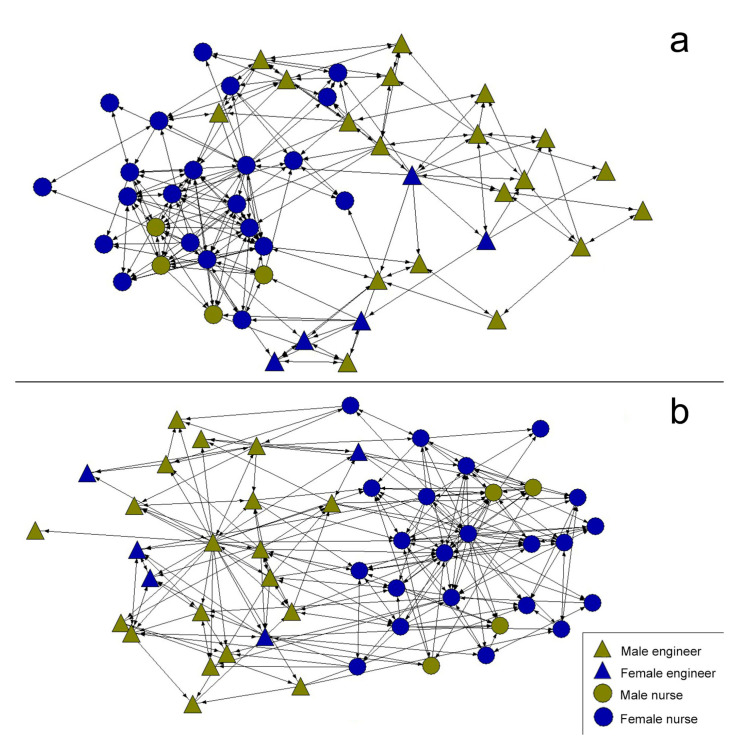
Post-intervention friendship network (**a**) and collaboration network (**b**).

**Figure 4 healthcare-08-00220-f004:**
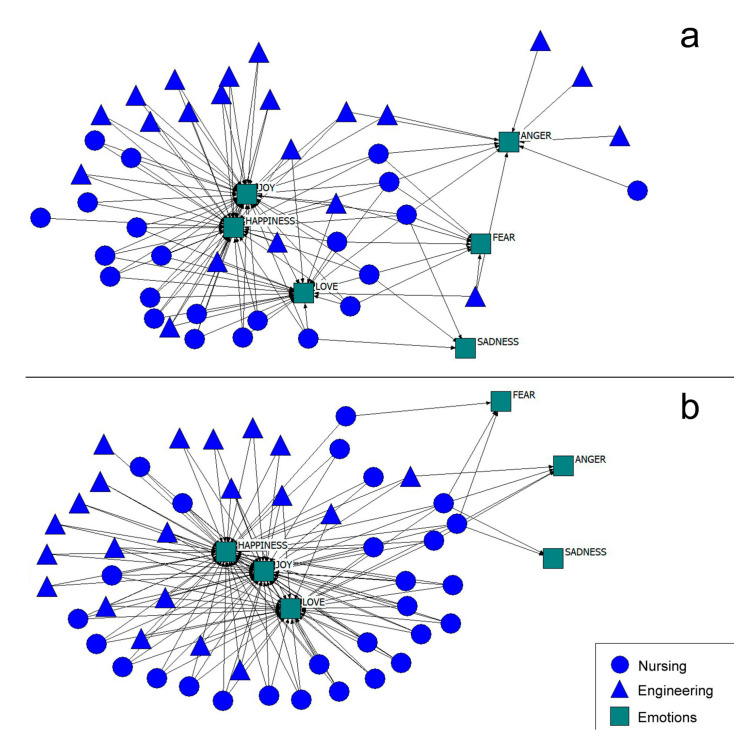
Post-intervention emotion network that are present (**a**) and that shoul be present (**b**).

**Figure 5 healthcare-08-00220-f005:**
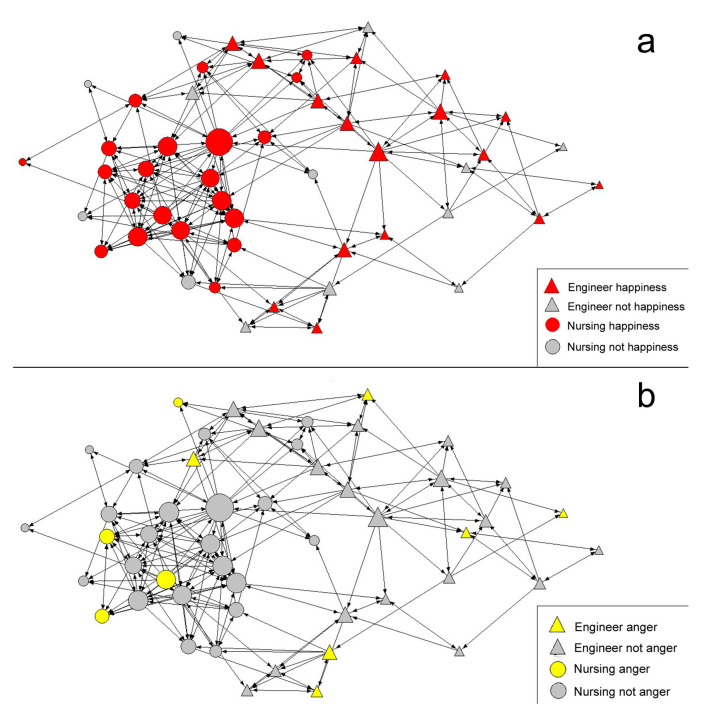
Graph of the post-intervention friendship network with Happiness emotion (**a**) and with Anger emotion (**b**).

**Table 1 healthcare-08-00220-t001:** Sample description.

Degree	Sex				TOTAL
	Men		Women		
	N	%	N	%	
Nursing	4	15.4	22	84.6	26
Computer engineering	19	79.2	5	22.08	24
TOTAL	23		27		50

**Table 2 healthcare-08-00220-t002:** Dichotomization of network interactions.

Network	Centrality Variable	Values
Collaboration	Without support	0, 1, 2
	With support	3, 4
Friendship	Whithout frienship	0, 1, 2
	With friendship	3, 4

**Table 3 healthcare-08-00220-t003:** Network Density.

Pre-Intervention		Post-Intervention	
Friendship	Collaboration	Friendship	Collaboration
0.074	0.053	0.105	0.107

**Table 4 healthcare-08-00220-t004:** Emotions pre-intervention.

Present Emotions											
Happiness		Love		Joy		Anger		Fear		Sadness	
Nur.	Eng.	Nur.	Eng.	Nur.	Eng.	Nur.	Eng.	Nur.	Eng.	Nur.	Eng.
61.5%	75%	31%	12.5%	61.5%	62.5%	27%	20.1%	27%	4.2%	3.9%	8.3%
**Emotions That Should Be Present**											
Happiness		Love		Joy		Anger		Fear		Sadness	
Nur.	Eng.	Nur.	Eng.	Nur.	Eng.	Nur.	Eng.	Nur.	Eng.	Nur.	Eng.
100%	91.7%	88.5%	45.8%	100%	95.8%	3.8%	4.2%	7.7%	0	3.8%	0

**Table 5 healthcare-08-00220-t005:** Emotions post-intervention.

Present Emotions											
Happiness		Love		Joy		Anger		Fear		Sadness	
Nur.	Eng.	Nur.	Eng.	Nur.	Eng.	Nur.	Eng.	Nur.	Eng.	Nur.	Eng.
80.8%	66.7%	61.5%	25%	77%	70.8%	15.4%	25%	23.1%	8.3%	11.5%	0
**Emotions That Should Be Present**											
Happiness		Love		Joy		Anger		Fear		Sadness	
Nur.	Eng.	Nur.	Eng.	Nur.	Eng.	Nur.	Eng.	Nur.	Eng.	Nur.	Eng.
96.2%	100%	84.6%	50%	92.3%	95.8%	11.5%	4.2%	11.5	0	7.7%	0

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
