# Peer review of "Networks and Emotions in Cooperative Work: A Quasi-Experimental Study in University Nursing and Computer Engineering Students"

_healthcare, 2020, doi:10.3390/healthcare8030220_

Round 1

Reviewer 1 Report

This paper presents a quasi-experimental study, addressing the interdisciplinary collaborative work between nursing and computer engineering students. The social-network analysis was applied.

The paper presents a novel research outcome but not groundbreaking. The research design sounds and the structure of the paper are very good.

However, a minor issue should be fixed. I would like to suggest the authors rewrite the introduction. The introduction provides a background of the research. But the aim of the research is not well presented which leads the reader a bit confused about the contribution of the paper. I recommend the authors to read lines 179-182 and make the introduction clearer.

Reviewer 2 Report

This paper aims at describing how the emotional relations constructed through collaborative networks between students impact the learning process and how interdisciplinary interventions can transform such networks.  Drawing on Human Relations Theory, the authors employ Social Network Analysis to describe the evolution of these social networks in relation to the emotional status expressed by students. The paper highlights the existence of specific relational patterns and put them in relation to various levels of academic performance. Higher levels of interaction are associated with higher performance.

I believe this article initiate an interesting discussion and proposes a viable and expandable approach to the study of educational environments. The increase in the density of friendship and collaboration networks in the post-intervention evaluation is definitively an interesting finding. However, the educational aspect of the research seems to be partially underdiscussed. This regard especially 1) the kind of the intervention and 2) the process of evaluation of academic performance. Indeed, in the first case different forms of intervention might lead to different outcomes (and not necessarily positive ones) and, in the second, the evaluation of academic performance can be influenced by many external factors. I believe authors should address these points and outline how the employed methodology was design to avoid bias.

Other comments:

Line 201 to 206 : The point made by the authors concerning the “bridging actors” is interesting but the proposed difference between Friendship and Collaboration networks is not easy to identify in the charts. (1a,1b)

Line 252 to 255: I suggest rephrasing the sentence as it is not clear, at a first sight, which kind of correlations is being discussed.

Line 331 to 336 and 360-361: I am not sure homophily is sufficient to explain this trait. In the methodology, authors state that students were selected  “in different campuses 113 kilometers apart”. Unlike each student involved in the project comes form a different campus, geographical/social proximity might have also influenced this result.

Line 338 to 339: the authors state that, given that students with higher grades had more intense social interactions, then “social interactions had an impact on their academic results”. Couldn’t it be the other way around?

Style Remakes

Sometimes sentences tend to be too long and with many subordinates. While grammatically correct, this might result in a lack of clarity. Accordingly, I would suggest rephrasing long sentences.

Reviewer 3 Report

  • Paper is written very clear and its focus is stringent. 
  • The conclusion need to be improved. There are repeating details about methods, results, interpretations and arguments.
  • The term "curried out" had repeated in many occasions. It will be a good if you can use similar term to avoid the repetition.
